# Sample Selection for Fair and Robust Training

**Yuji Roh**
KAIST
yuji.roh@kaist.ac.kr

**Kangwook Lee**
University of Wisconsin-Madison
kangwook.lee@wisc.edu

**Steven Euijong Whang**[*]
KAIST
swhang@kaist.ac.kr

**Changho Suh**
KAIST
chsuh@kaist.ac.kr

## Abstract

Fairness and robustness are critical elements of Trustworthy AI that need to be addressed together. Fairness is about learning an unbiased model while robustness is about learning from corrupted data, and it is known that addressing only one of them may have an adverse affect on the other. In this work, we propose a sample selection-based algorithm for fair and robust training. To this end, we formulate a combinatorial optimization problem for the unbiased selection of samples in the presence of data corruption. Observing that solving this optimization problem is strongly NP-hard, we propose a greedy algorithm that is efficient and effective in practice. Experiments show that our algorithm obtains fairness and robustness that are better than or comparable to the state-of-the-art technique, both on synthetic and benchmark real datasets. Moreover, unlike other fair and robust training baselines, our algorithm can be used by only modifying the sampling step in batch selection without changing the training algorithm or leveraging additional clean data.

## 1 Introduction

Trustworthy AI is becoming essential for modern machine learning. While a traditional focus of machine learning is to train the most accurate model possible, companies that actually deploy AI including Microsoft [2021], Google [2020], and IBM [2020] are now pledging to make their AI systems fair, robust, interpretable, and transparent as well. Among the key objectives, we focus on fairness and robustness because they are closely related and need to be addressed together. Fairness is about learning an unbiased model while robustness is about learning a resilient model even on noisy data, and both issues root from the same training data. A recent work [Roh et al., 2020] shows that improving fairness while ignoring noisy data may lead to a worse accuracy-fairness tradeoff. Likewise, focusing on robustness only may have an adverse affect on fairness as we explain below.

There are several possible approaches for supporting fairness and robustness together. One is an in-processing approach where the model architecture itself is modified to optimize the two objectives. In particular, the state-of-the-art approach called FR-Train [Roh et al., 2020] uses adversarial learning to train a classifier and two discriminators for fairness and robustness. However, such an approach requires the application developer to use a specific model architecture and is thus restrictive. Another approach is to preprocess the data [Kamiran and Calders, 2011] and remove any bias and noise before model training, but this requires modifying the data itself. In addition, we are not aware of a general data cleaning method that is designed to explicitly improve model fairness and robustness together.

Instead, we propose an adaptive sample selection approach for the purpose of improving fairness and robustness. This approach can easily be deployed as a batch selection method, which does not require

---

[*]Corresponding author

35th Conference on Neural Information Processing Systems (NeurIPS 2021).

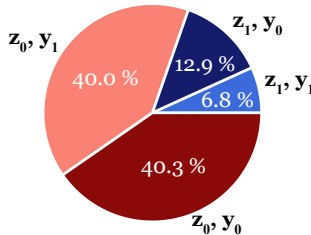
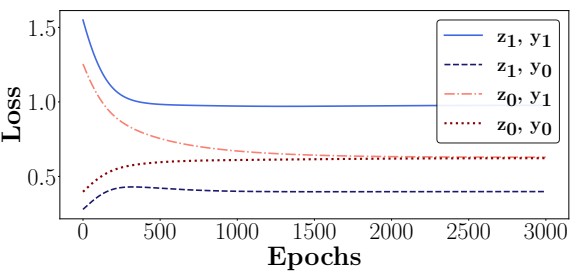

(a) The size ratios of (label y, sensitive group z) sets within the COMPAS dataset.

(b) Tracing the model losses on (label y, sensitive group z) sets when training logistic regression on the COMPAS dataset.

Figure 1: The ProPublica COMPAS dataset [Angwin et al., 2016] has bias where the set $(y = 1, z = 1)$ has the smallest size in Figure 1a. However, when training a logistic regression model on this dataset, the same set has the highest loss at every epoch as shown in Figure 1b. Blindly applying clean selection at any point will thus discard $(y = 1, z = 1)$ samples the most, which results in even worse bias and possibly worse fairness.

modifying the model or data. Our techniques build on top of two recent lines of sample selection research: clean selection for robust training [Rousseeuw, 1984, Song et al., 2019, Shen and Sanghavi, 2019] and batch selection for fairness [Roh et al., 2021]. Clean selection is a standard approach for discarding samples that have high-loss values and are thus considered noisy. More recently, batch selection for fairness has been proposed where the idea is to adaptively adjust sampling ratios among sensitive groups (e.g., black or white populations) so that the trained model is not discriminative.

We note that state-of-the-art clean selection techniques like Iterative Trimmed Loss Minimization (ITLM) [Shen and Sanghavi, 2019] are not designed to address fairness and may actually worsen the data bias when used alone. For example, Figure 1 shows how the ProPublica COMPAS dataset (used by U.S. courts to predict criminal recidivism rates [Angwin et al., 2016]) can be divided into four sets where the label y is either 0 or 1, and the sensitive attribute z is either 0 or 1. The sizes of the sets are biased where $(y = 1, z = 1)$ is the smallest. However, the model loss on $(y = 1, z = 1)$ is the highest among the sets as shown in Figure 1b. Hence, blindly applying clean selection to remove high-loss samples will discard $(y = 1, z = 1)$ samples the most, resulting in a worse bias that may negatively affect fairness as we demonstrate in Sec. 2.

We thus formulate a combinatorial optimization problem for fair and robust sample selection that achieves unbiased sampling in the presence of data corruption. To avoid discrimination on a certain group, we adaptively adjust the maximum number of samples per (y, z) set using the recently proposed FairBatch system [Roh et al., 2021], which solves a bilevel optimization problem of unfairness mitigation and standard empirical risk minimization through batch selection. We show that our optimization problem for sample selection can be viewed as a multidimensional knapsack problem and is thus strongly NP-hard [Garey and Johnson, 1979]. We then propose an efficient greedy algorithm that is effective in practice. Our method supports prominent group fairness measures: equalized odds [Hardt et al., 2016] and demographic parity [Feldman et al., 2015]. For data corruption, we currently assume noisy labels [Song et al., 2020], which can be produced by random flipping or adversarial attacks.

Experiments on synthetic and benchmark real datasets (COMPAS [Angwin et al., 2016] and Adult-Census [Kohavi, 1996]) show that our method obtains accuracy and fairness results that are better than baselines using fairness algorithms [Zafar et al., 2017a,b, Roh et al., 2021] and on par with and sometimes better than FR-Train [Roh et al., 2020], which leverages additional clean data.

**Notation** Let $\theta$ be the model weights, $x \in \mathbb{X}$ be the input feature to the classifier, $y \in \mathbb{Y}$ be the true class, and $\hat{y} \in \mathbb{Y}$ be the predicted class where $\hat{y}$ is a function of $(x, \theta)$. Let $z \in \mathbb{Z}$ be a sensitive attribute, e.g., race or gender. Let $d = (x, y)$ be a training sample that contains a feature and label. Let $S$ be a selected subset from the entire dataset $D$, where the cardinality of $D$ is $n$. In the model training, we use a loss function $\ell_\theta(\cdot)$ where a smaller value indicates a more accurate prediction.

## 2 Unfairness in Clean Selection

We demonstrate how clean selection that only focuses on robust training may lead to unfair models. Throughout this paper, we use ITLM [Shen and Sanghavi, 2019] as a representative clean selection

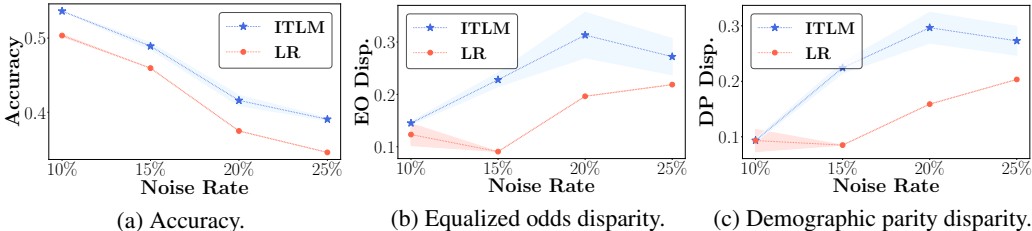

(a) Accuracy.  (b) Equalized odds disparity.  (c) Demographic parity disparity.

Figure 2: Performances of logistic regression with and without ITLM (ITLM and LR, respectively) on the ProPublica COMPAS dataset while varying the noise rate using label flipping [Paudice et al., 2018]. Although ITLM improves accuracy (higher is better), the fairness worsens according to the equalized odds and demographic parity disparity measures (lower is better).

method for its simplicity, although other methods can be used as well. We first introduce ITLM and demonstrate how ITLM can actually worsen fairness.

**ITLM**  Cleaning samples based on their loss values is a traditional robust training technique [Rousseeuw, 1984] that is widely used due to its efficiency. Recently, ITLM was proposed as a scalable approach for sample selection where lower loss samples are considered cleaner. The following optimization problem is solved with theoretical guarantees of convergence:

$$\min_{S:|S|=\lfloor \tau n \rfloor} \sum_{s_i \in S} \ell_\theta(s_i)$$

where $S$ is the selected set, $s_i$ is a sample in $S$, and $\tau$ is the ratio of clean samples in the entire data.

**Potential Unfairness**  We show how applying ITLM may lead to unfairness through a simple experiment. We use the benchmark ProPublica COMPAS dataset [Angwin et al., 2016] where the label y indicates recidivism, and the sensitive attribute z is set to gender. We inject noise into the training set using label flipping techniques [Paudice et al., 2018]. For simplicity, we assume that ITLM knows the clean ratio $\tau$ of the data and selects $\lfloor \tau n \rfloor$ samples. We train a logistic regression model on the dataset, and ITLM iteratively selects clean samples during the training.

We show the accuracy and fairness performances of logistic regression with and without ITLM while varying the noise rate (i.e., portion of labels flipped) in Figure 2. As expected, ITLM increasingly improves the model accuracy for larger noise rates as shown in Figure 2a. At the same time, however, the fairness worsens according to the two measures we use in this paper (see definitions in Sec. 5) – equalized odds [Hardt et al., 2016] disparity and demographic parity [Feldman et al., 2015] disparity – as shown in Figures 2b and 2c, respectively. The results confirm our claims in Sec. 1.

## 3   Framework

We provide an optimization framework for training models on top of clean selection with fairness constraints. The entire optimization involves three joint optimizations, which we explain in a piecewise fashion: (1) how to perform clean selection with fairness constraints, (2) how to train a model on top of the selected samples, and (3) how to generate the fairness constraints. In Sec. 4 we present a concrete algorithm for solving all the optimizations.

### 3.1   Clean Selection with Fairness Constraints

The first optimization is to select clean samples such that the loss is minimized while fairness constraints are satisfied. For now, we assume that the fairness constraints are given where the selected samples of a certain set (y = $y$, z = $z$) cannot exceed a portion $\lambda_{(y,z)}$ of the selected samples of the class y = $y$. Notice that $\sum_z \lambda_{(y,z)} = 1, \forall y \in \mathbb{Y}$. The fairness constraints are specified as upper bounds because using equality constraints may lead to infeasible solutions. We explain how the $\lambda$

values are determined in Sec. 3.3. We thus solve the following optimization for sample selection:

$$\min_{\mathrm{p}} \sum_{i=1}^{n} \ell_\theta(d_i)\, p_i \tag{1}$$

$$\text{s.t. } \sum_{i=1}^{n} p_i \leq \tau n\,, \quad \sum_{j \in \mathbb{I}_{(y,z)}} p_j \leq \lambda_{(y,z)} |S_y|,\ \forall (y,z) \in \mathbb{Y} \times \mathbb{Z}, \tag{2}$$

$$p_i \in \{0,1\},\ i = 1,...,n$$

where $p_i$ indicates whether the data sample $d_i$ is selected or not, $\mathbb{I}_{(y,z)}$ is an index set of the $(y,z)$ class, and $S_y$ is the selected samples for y $= y$. Note that $|S_y| = \sum_{j \in \mathbb{I}_y} p_j$. In Eq. 2, the first constraint comes from ITLM while the second constraint is for the fairness.

An important detail is that we use inequalities in all of our constraints, unlike ITLM. The reason is that we may not be able to find a feasible solution of selected samples if we only use equality constraints. For example, if $\lambda_{(1,0)} = \lambda_{(1,1)} = 0.5$, but we have zero (y $= 1$, z $= 0$) samples, then the fairness equality constraint for (y $= 1$, z $= 0$) can never be satisfied.

## 3.2 Model Training

The next optimization is to train a model on top of the selected samples from Sec. 3.1. Recall that the sample selection optimization only uses inequality constraints, which means that for some set (y $= y$, z $= z$), its size $|S_{(y,z)}|$ may be smaller than $\lambda_{(y,z)}|S_y|$. However, we would like to train the model as if there were $\lambda_{(y,z)}|S_y|$ samples for the intended fairness performance. We thus reweight each set (y $= y$, z $= z$) by $\frac{\lambda_{(y,z)}|S_y|}{|S_{(y,z)}|}$ and perform weighted empirical risk minimization (ERM) for model training. We then solve the bilevel optimization of model training and sample selection in Sec. 3.1 as follows:

$$\min_{\theta} \sum_{s_i \in S_{\boldsymbol{\lambda}}} \sum_{(y,z) \in (\mathbb{Y}, \mathbb{Z})} \mathbb{I}_{S_{(y,z)}}(s_i)\, \frac{\lambda_{(y,z)}|S_y|}{|S_{(y,z)}|} \ell_\theta(s_i) \tag{3}$$

$$S_{\boldsymbol{\lambda}} = \underset{S = \{d_i | p_i = 1\}: \text{Eq. 2}}{\arg\min} \quad \text{Eq. 1} \tag{4}$$

where $\boldsymbol{\lambda}$ is the set of $\lambda_{(y,z)}$ for all $z \in \mathbb{Z}$, $y \in \mathbb{Y}$.

## 3.3 Fairness Constraint Generation

To determine the $\lambda_{y,z}$ values, we adaptively adjust sampling ratios of sensitive groups using the techniques in FairBatch [Roh et al., 2021]. For each batch selection, we compute the fairness of the intermediate model and decide which sensitive groups should be sampled more (or less) in the next batch. The sampling rates become the $\lambda_{(y,z)}$ values used in the fairness constraints of Eq. 2.

FairBatch supports all the group fairness measures used in this paper. As an illustration, we state FairBatch's optimization when using equalized odds disparity (see Sec. A.1 for the optimization using demographic parity disparity):

$$\min_{\boldsymbol{\lambda}} \max\{|L_{(y,z')} - L_{(y,z)}|\},\ z \neq z',\ z, z' \in \mathbb{Z},\ y \in \mathbb{Y} \tag{5}$$

where $L_{(y,z)} = 1/|S_{\boldsymbol{\lambda}(y,z)}| \sum_{S_{\boldsymbol{\lambda}(y,z)}} \ell_\theta(s_i)$, and $S_{\boldsymbol{\lambda}(y,z)}$ is a subset of $S_{\boldsymbol{\lambda}}$ from Eq. 4 for the (y $= y$, z $= z$) set. FairBatch adjusts $\boldsymbol{\lambda}$ in each epoch via a signed gradient-based algorithm to the direction of reducing unfairness. For example, if the specific sensitive group $z$ is less accurate on the y $= y$ samples in the epoch $t$ (i.e., $z$ is discriminated in terms of equalized odds), FairBatch selects more data from the (y $= y$, z $= z$) set to increase the model's accuracy on that set:

$$\lambda_{(y,z)}^{(t+1)} = \lambda_{(y,z)}^{(t)} - \alpha \cdot \text{sign}(L_{(y,z')} - L_{(y,z)}),\ z \neq z' \tag{6}$$

where $\alpha$ is the step size of $\lambda$ updates. To integrate FairBatch with our framework, we adaptively adjust $\boldsymbol{\lambda}$ as above for an intermediate model trained on the selected set $S_{\boldsymbol{\lambda}}$ from Eq. 4.

| **Algorithm 1:** Greedy-Based Clean and Fair Sample Selection | **Algorithm 2:** Overall Fair and Robust Model Training |
|---|---|
| **Input:** loss $\ell_\theta(\boldsymbol{d})$, fair ratio lambda $\boldsymbol{\lambda}$, clean ratio $\tau$ | **Input:** train data $(x_{\text{train}}, y_{\text{train}})$, clean ratio $\tau$, loss function $\ell_\theta(\cdot)$ |

**Algorithm 1:** Greedy-Based Clean and Fair Sample Selection

**Input:** loss $\ell_\theta(\boldsymbol{d})$, fair ratio lambda $\boldsymbol{\lambda}$, clean ratio $\tau$

profit $= \max(\ell_\theta(\boldsymbol{d})) - \ell_\theta(\boldsymbol{d})$
sortIdx $= \texttt{Sort}(\text{profit})$
$S \leftarrow [\,]$

**for** idx in sortIdx **do**
  **if** $d_{\text{idx}}$ does not violate any constraint in Eq. 8 w.r.t. $\boldsymbol{\lambda}$ and $\tau$ **then**
    Append $d_{\text{idx}}$ to $S$
**Output :** $S$

**Algorithm 2:** Overall Fair and Robust Model Training

**Input:** train data $(x_{\text{train}}, y_{\text{train}})$, clean ratio $\tau$, loss function $\ell_\theta(\cdot)$
$\boldsymbol{d} \leftarrow (x_{\text{train}}, y_{\text{train}})$
$\theta \leftarrow$ initial model parameters
$\boldsymbol{\lambda} = \{\lambda_{y,z} | y \in \mathbb{Y}, z \in \mathbb{Z}\} \leftarrow$ random sampling ratios
**for** *each epoch* **do**
  $S = \texttt{Algorithm1}(\ell_\theta(\boldsymbol{d}), \boldsymbol{\lambda}, \tau_y)$
  Draw minibatches from $S$ w.r.t. $\lambda_{y,z}|S_y|/|S_{(y,z)}|$
  **for** *each minibatch* **do**
    Update model parameters $\theta$ according to the minibatch
  Update $\forall \lambda_{y,z} \in \boldsymbol{\lambda}$ using the update rule as in Eq. 6
**Output :** model parameters $\theta$

## 4 Algorithm

We present our algorithm for solving the optimization problem in Sec. 3. We first explain how the clean selection with fairness constraints problem in Sec. 3.1 can be converted to a multi-dimensional knapsack problem, which has known solutions. We then describe the full algorithm that includes the three functionalities: clean and fair selection, model training, and fairness constraint generation.

The clean and fair selection problem can be converted to an equivalent multi-dimensional knapsack problem by (1) maximizing the sum of $(\max(\ell_\theta(\boldsymbol{d})) - \ell_\theta(d_i))$'s instead of minimizing the sum of $\ell_\theta(d_i)$'s where $\boldsymbol{d}$ is the set of $d_i$'s and (2) re-arranging the fairness constraints so that the right-hand side expressions become constants (instead of containing the variable $S_y$) as follows:

$$\max \sum_{i=1}^{n} (\max(\ell_\theta(\boldsymbol{d})) - \ell_\theta(d_i))\, p_i \tag{7}$$

$$\text{s.t.} \sum_{i=1}^{n} p_i \leq \tau n, \;\; \sum_{i=1}^{n} w_i p_i \leq \tau n, \text{ where } w_i = \begin{cases} 1 & \text{if } d_i \notin D_y \\ 1 - \lambda_{(y,z)} & \text{if } d_i \in D_y \text{ and } d_i \notin D_z \\ 2 - \lambda_{(y,z)} & \text{if } d_i \in D_{(y,z)} \end{cases}, \; \forall (y,z) \in \mathbb{Y} \times \mathbb{Z}, \tag{8}$$

$$p_i \in \{0, 1\}, \; i = 1, ..., n$$

where $D_{(y,z)}$ is a subset for the $(y, z)$ class. More details on the conversion are in Sec. A.2.

The multi-dimensional knapsack problem is known to be strongly NP-hard [Garey and Johnson, 1979]. Although exact algorithms have been proposed, they have exponential time complexities [Kellerer et al., 2004] and are thus impractical. There is a polynomial-time approximation scheme (PTAS) [Caprara et al., 2000], but the actual computation time increases exponentially for more accurate solutions [Kellerer et al., 2004]. Since our method runs within a single model training, we cannot tolerate such runtimes and thus opt for a greedy algorithm that is efficient and known to return reasonable solutions in practice [Akçay et al., 2007]. Algorithm 1 takes a greedy approach by sorting all the samples in descending order by their $(\max(\ell_\theta(\boldsymbol{d})) - \ell_\theta(d_i))$ values and then sequentially selecting the samples that do not violate any of the constraints. The computational complexity is thus $\mathcal{O}(n \log n)$ where $n$ is the total number of samples.

We now present our overall model training process for weighted ERM in Algorithm 2. For each epoch, we first select a clean and fair sample set $S$ using Algorithm 1. We then draw minibatches from $S$ according to the $(\mathrm{y} = y, \mathrm{z} = z)$-wise weights described in Sec. 3.2 and update the model parameters $\theta$ via stochastic gradient descent. A minibatch selection with uniform sampling can be considered as an unbiased estimator of the ERM, so sampling each $(\mathrm{y} = y, \mathrm{z} = z)$ set in proportion to its weight results in an unbiased estimator of the weighted ERM. Finally, at the end of the epoch, we update the $\boldsymbol{\lambda}$ values using the update rules as in Eq. 6. We discuss about convergence in Sec. A.3.

# 5 Experiments

We evaluate our proposed algorithm. We use logistic regression for all experiments. We evaluate our models on separate clean test sets and repeat all experiments with 5 different random seeds. We use PyTorch, and all experiments are run on Intel Xeon Silver 4210R CPUs and NVIDIA Quadro RTX 8000 GPUs. More detailed settings are in Sec. B.1.

**Fairness Measures**   We focus on two representative group fairness measures: (1) equalized odds (EO) [Hardt et al., 2016], whose goal is to obtain the same accuracy between sensitive groups conditioned on the true labels and (2) demographic parity (DP) [Feldman et al., 2015], which is satisfied when the sensitive groups have the same positive prediction ratio. We evaluate the fairness disparities over sensitive groups as follows: *EO disparity* $= \max_{z \in \mathbb{Z}, y \in \mathbb{Y}} |\Pr(\hat{y} = 1 | z = z, y = y) - \Pr(\hat{y} = 1 | y = y)|$ and *DP disparity* $= \max_{z \in \mathbb{Z}} |\Pr(\hat{y} = 1 | z = z) - \Pr(\hat{y} = 1)|$. Note that the classifier is perfectly fair when the fairness disparity is zero.

**Datasets**   We use a total of three datasets: one synthetic dataset and two real benchmark datasets. We generate a synthetic dataset using a similar method to Zafar et al. [2017a]. The synthetic dataset has 3,200 samples and consists of two non-sensitive features $(x_1, x_2)$, one sensitive feature z, and one label class y. Each sample $(x_1, x_2, y)$ is drawn from the following Gaussian distributions: $(x_1, x_2) | y = 1 \sim \mathcal{N}([1; 1], [5, 1; 1, 5])$ and $(x_1, x_2) | y = 0 \sim \mathcal{N}([-1; -1], [10, 1; 1, 3])$. We add the sensitive feature z to have a biased distribution: $\Pr(z = 1) = 7 \Pr((x_1', x_2') | y = 1) / [7 \Pr((x_1', x_2') | y = 1) + \Pr((x_1', x_2') | y = 0)]$ where $(x_1', x_2') = (x_1 \cos(\pi/5) - x_2 \sin(\pi/5), x_1 \sin(\pi/5) + x_2 \cos(\pi/5))$. We visualize the synthetic dataset in Sec. B.2. We utilize two real datasets, ProPublica COMPAS [Angwin et al., 2016] and AdultCensus [Kohavi, 1996], and use the same pre-processing in IBM Fairness 360 [Bellamy et al., 2019]. COMPAS and AdultCensus consist of 5,278 and 43,131 samples, respectively. The labels in COMPAS indicate recidivism, and the labels in AdultCensus indicate each customer's annual income level. For both datasets, we use gender as the sensitive attribute. Our experiments do not use any direct personal identifier, such as name or date of birth.

**Noise Injection**   We use two methods for noise injection: (1) label flipping [Paudice et al., 2018], which minimizes the model accuracy (Label Flipping) and (2) targeted label flipping [Roh et al., 2020], which flips the labels of a specific group (Group-Targeted Label Flipping). When targeting a group in (2), we choose the one that, when attacked, results in the lowest model accuracy on all groups. The two methods represent different scenarios for label flipping attacks. In Secs. 5.1, 5.2, and 5.4, we flip 10% of labels in the training data, and in Sec. 5.3, we flip 10% to 20% of labels (i.e., varying the noise rate) in the training data.

**Baselines**   We compare our proposed algorithm with four types of baselines: (1) *vanilla* training using logistic regression (LR); (2) *robust only* training using ITLM [Shen and Sanghavi, 2019]; (3) *fair only* training using FairBatch [Roh et al., 2021]; and (4) *fair and robust* training where we evaluate two baselines and a state-of-the-art method called FR-Train [Roh et al., 2020]. The two baselines are as follows where each runs in two steps:

- ITLM→FB: Runs ITLM and then FairBatch on the resulting clean samples.
- ITLM→Penalty: Runs ITLM and then an in-processing fairness algorithm [Zafar et al., 2017a,b] on the clean samples. The fairness algorithm adds a penalty term to the loss function for reducing the covariance between the sensitive attribute and the predicted labels.

FR-Train [Roh et al., 2020] performs adversarial training between a classifier and fair and robust discriminators. For the robust training, FR-Train relies on a separate clean validation set. Since FR-Train benefits from additional data, its performances can be viewed as an upper bound for our algorithm, which does not utilize such data.

In Sec. B.3, we also compare with other two-step baselines that improve the fairness both during and after clean sample selection.

**Hyperparameters**   We choose the step size for updating $\lambda$ (i.e., $\alpha$ in Eq. 6) within the candidate set {0.0001, 0.0005, 0.001} using cross-validation. We assume the clean ratio $\tau$ is known for any dataset. If the clean ratio is not known in advance, it can be inferred using cross-validation [Liu and Tao, 2015, Yu et al., 2018]. For all baselines, we start from a candidate set of hyperparameters and

Table 1: Performances on the *synthetic* test set w.r.t. equalized odds disparity (EO Disp.) and demographic parity disparity (DP Disp.). We compare our algorithm with four types of baselines: (1) vanilla training: LR; (2) robust training: ITLM [Shen and Sanghavi, 2019]; (3) fair training: FB [Roh et al., 2021]; and (4) fair and robust training: ITLM→FB, ITLM→Penalty [Zafar et al., 2017a,b], and FR-Train [Roh et al., 2020]. We flip 10% of labels in the training data. Experiments are repeated 5 times. We highlight the best and second-best performances among the fair and robust algorithms.

| Method | Label Flipping | | | | Group-Targeted Label Flipping | | | |
| --- | --- | --- | --- | --- | --- | --- | --- | --- |
| | Acc. | EO Disp. | Acc. | DP Disp. | Acc. | EO Disp. | Acc. | DP Disp. |
| LR | .665±.003 | .557±.015 | .665±.003 | .400±.010 | .600±.002 | .405±.008 | .600±.002 | .300±.006 |
| ITLM | .716±.001 | .424±.003 | .716±.001 | .380±.002 | .719±.001 | .315±.017 | .719±.001 | .287±.009 |
| FB | .509±.017 | .051±.018 | .683±.002 | .063±.019 | .544±.005 | .072±.010 | .526±.003 | .082±.002 |
| ITLM→FB | .718±.003 | .199±.020 | **.725±.002** | .089±.032 | .707±.001 | .108±.030 | .704±.003 | .067±.027 |
| ITLM→Penalty | .651±.051 | .172±.046 | .674±.012 | .068±.014 | .706±.001 | .080±.004 | .688±.004 | **.044±.004** |
| **Ours** | **.727±.002** | **.064±.005** | .720±.001 | **.006±.001** | **.726±.001** | **.040±.002** | **.720±.001** | **.039±.007** |
| FR-Train | **.722±.005** | **.078±.010** | .711±.003 | **.057±.027** | **.715±.003** | **.046±.011** | **.706±.011** | .079±.030 |

Table 2: Performances on the *COMPAS* test set w.r.t. equalized odds disparity (EO Disp.) and demographic parity disparity (DP Disp.). Other experimental settings are identical to Table 1.

| Method | Label Flipping | | | | Group-Targeted Label Flipping | | | |
| --- | --- | --- | --- | --- | --- | --- | --- | --- |
| | Acc. | EO Disp. | Acc. | DP Disp. | Acc. | EO Disp. | Acc. | DP Disp. |
| LR | .503±.002 | .123±.021 | .503±.002 | .093±.020 | .513±.000 | .668±.000 | .513±.000 | .648±.000 |
| ITLM | .536±.000 | .145±.003 | .536±.000 | .094±.003 | .539±.002 | .573±.019 | .539±.002 | .547±.015 |
| FB | .509±.001 | .083±.013 | .503±.002 | .046±.010 | .525±.004 | .093±.003 | .507±.004 | .059±.010 |
| ITLM→FB | .520±.001 | **.064±.013** | .523±.003 | **.039±.009** | .538±.008 | .315±.074 | .520±.013 | .130±.105 |
| ITLM→Penalty | **.523±.003** | .074±.027 | .525±.001 | .059±.008 | .517±.014 | .436±.028 | .514±.007 | .094±.022 |
| **Ours** | .521±.000 | **.044±.000** | **.544±.000** | **.025±.000** | **.545±.009** | **.084±.012** | **.521±.000** | **.024±.000** |
| FR-Train | **.620±.009** | .074±.015 | **.607±.016** | .050±.013 | **.611±.029** | **.081±.020** | **.597±.039** | **.045±.014** |

use cross-validation to choose the hyperparameters that result in the best fairness while having an accuracy that best aligns with other results. More hyperparameters are described in Sec. B.1.

## 5.1 Accuracy and Fairness

We first compare the accuracy and fairness results of our algorithm with other methods on the synthetic dataset in Table 1. We inject noise into the synthetic dataset either using label flipping or targeted label flipping. For both cases, our algorithm achieves the highest accuracy and fairness results compared to the baselines. Compared to LR, our algorithm has higher accuracy and lower EO and DP disparities, which indicates better fairness. Compared to ITLM and FB, our algorithm shows much better fairness and accuracy, respectively. FB's accuracy is noticeably low as a result of a worse accuracy-fairness tradeoff in the presence of noise. We now compare with the fair and robust training methods. Both ITLM→FB and ITLM→Penalty usually show worse accuracy and fairness compared to our algorithm, which shows that noise and bias cannot easily be mitigated in separate steps.

Compared to FR-Train, our algorithm surprisingly has similar accuracy and better fairness, which suggests that sample selection techniques like ours can outperform in-processing techniques like FR-Train that also rely on clean validation sets. We also perform an accuracy-fairness trade-off comparison between our algorithm and FR-Train in Sec. B.4, where the trends are consistent with the results in Table 1.

We now make the same comparisons using real datasets. We show the COMPAS dataset results here in Table 2 and the AdultCensus dataset results in Sec. B.5 as they are similar. The results for LR, ITLM, and FB are similar to those for the synthetic dataset. Compared to the fair and robust algorithms ITLM->FB, ITLM->Penalty, and FR-Train, our algorithm usually has the best or second-best accuracy and fairness values as highlighted in Table 2. Unlike in the synthetic dataset, FR-Train obtains very high accuracy values with competitive fairness. We suspect that FR-Train is benefiting from its clean validation set, which motivates us to perform a more detailed comparison with our algorithm in the next section.

Table 3: Detailed comparison with FR-Train on the COMPAS test set using 10% label flipping. We use the following three different validation sets for FR-Train: (1) noisy; (2) noisy, but cleaned with ITLM; and (3) clean.

| Method | Acc. | EO Disp. | Acc. | DP Disp. |
|---|---|---|---|---|
| FR-Train with clean val. set | .620±.009 | .074±.015 | .607±.016 | .050±.013 |
| FR-Train with noisy val. set cleaned with ITLM | .531±.033 | .087±.018 | .530±.024 | .059±.018 |
| FR-Train with noisy val. set | .502±.034 | .073±.021 | .513±.035 | .041±.024 |
| **Ours** | .521±.000 | .044±.000 | .544±.000 | .025±.000 |

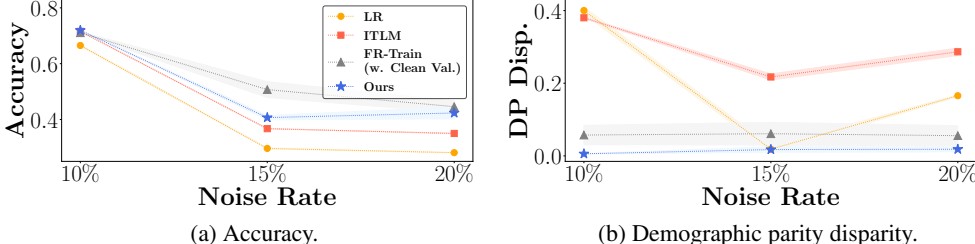

(a) Accuracy.  (b) Demographic parity disparity.

Figure 3: Performances of LR, ITLM, FR-Train, and our algorithm (Ours) on the synthetic data while varying the noise rate using label flipping [Paudice et al., 2018].

## 5.2 Detailed Comparison with FR-Train

We perform a more detailed comparison between our algorithm and FR-Train. In Sec. 5.1, we observed that FR-Train takes advantage of its clean validation set to obtain very high accuracy values on the COMPAS dataset. We would like to see how that result changes when using validation sets that are less clean. In Table 3, we compare with two scenarios: (1) FR-Train has no clean data, so the validation set is just as noisy as the training set with 10% of its labels flipped and (2) the validation set starts as noisy, but is then cleaned using ITLM, which is the best we can do if there is no clean data. As a result, our algorithm is comparable to the realistic scenario (2) for FR-Train.

We also compare runtimes (i.e., wall clock times in seconds) of our algorithm and FR-Train on the synthetic and COMPAS datasets w.r.t. equalized odds. As a result, Table 4 shows that our algorithm is faster on the synthetic dataset, but a bit slower on the COMPAS dataset. Our algorithm's runtime bottleneck is the greedy algorithm (Algorithm 1) for batch selection. FR-Train trains slowly because it trains three models (one classifier and two discriminators) together. Hence, there is no clear winner.

Table 4: Runtime comparison (in seconds) of our algorithm and FR-Train on the synthetic and COMPAS test sets w.r.t. equalized odds disparity. Other experimental settings are identical to Table 1.

| Method | Synthetic | COMPAS |
|---|---|---|
| FR-Train | 90.986 | 230.304 |
| **Ours** | 44.746 | 241.291 |

## 5.3 Varying the Noise Rate

We compare the algorithm performances when varying the noise rate (i.e., portion of labels flipped) in the data. Figure 3 shows the accuracy and fairness (w.r.t. demographic parity disparity) of LR, ITLM, FR-Train, and our algorithm when varying the noise rate in the synthetic data. Even if the noise rate increases, the relative performances among the four methods do not change significantly. Our algorithm still outperforms LR and ITLM in terms of accuracy and fairness and is comparable to FR-Train having worse accuracy, but better fairness. The results w.r.t. equalized odds disparity are similar and can be found in Sec. B.6.

## 5.4 Ablation Study

In Table 5, we conduct an ablation study to investigate the effect of each component in our algorithm. We consider three ablation scenarios: (1) remove the fairness constraints in Eq. 2, which reduce discrimination in sample selection, (2) remove the weights in Eq. 3, which ensure fair model training, and (3) remove both functionalities (i.e., same as ITLM). As a result, each ablation scenario leads

Table 5: Ablation study on the synthetic and COMPAS test sets using 10% label flipping. We consider the following three ablation scenarios: (1) w/o fairness constraints; (2) w/o ERM weights; and (3) w/o both.

| Method | Synthetic | | | | COMPAS | | | |
|---|---|---|---|---|---|---|---|---|
| | Acc. | EO Disp. | Acc. | DP Disp. | Acc. | EO Disp. | Acc. | DP Disp. |
| W/o both | .716±.001 | .424±.003 | .716±.009 | .380±.002 | .536±.000 | .145±.003 | .536±.000 | .094±.003 |
| W/o fairness const. | .718±.002 | .070±.004 | .723±.001 | .036±.002 | .520±.002 | .110±.005 | .520±.002 | .073±.002 |
| W/o ERM weights | .727±.001 | .233±.014 | .727±.000 | .251±.009 | .531±.004 | .152±.018 | .529±.007 | .058±.020 |
| **Ours** | .727±.002 | .064±.005 | .720±.001 | .006±.001 | .521±.000 | .044±.000 | .544±.000 | .025±.000 |

to either worse accuracy and fairness or slightly-better accuracy, but much worse fairness. We thus conclude that both functionalities are necessary.

# 6   Related Work

**Fair & Robust Training**   We cover the literature for fair training, robust training, and fair and robust training, in that order.

For model fairness, many definitions have been proposed to address legal and social issues [Narayanan, 2018]. Among the definitions, we focus on group fairness: equalized odds [Hardt et al., 2016] and demographic parity [Feldman et al., 2015]. The algorithms for obtaining group fairness can be categorized as follows: (1) pre-processing techniques that minimize bias in data [Kamiran and Calders, 2011, Zemel et al., 2013, Feldman et al., 2015, du Pin Calmon et al., 2017, Choi et al., 2020, Jiang and Nachum, 2020], (2) in-processing techniques that revise the training process to prevent the model from learning the bias [Kamishima et al., 2012, Zafar et al., 2017a,b, Agarwal et al., 2018, Zhang et al., 2018, Cotter et al., 2019], and (3) post-processing techniques that modify the outputs of the trained model to ensure fairness [Kamiran et al., 2012, Hardt et al., 2016, Pleiss et al., 2017, Chzhen et al., 2019]. However, these fairness-only approaches do not address robust training as we do. Beyond group fairness, there are other important fairness measures including individual fairness [Dwork et al., 2012] and causality-based fairness [Kilbertus et al., 2017, Kusner et al., 2017, Zhang and Bareinboim, 2018, Nabi and Shpitser, 2018, Khademi et al., 2019]. Extending our algorithm to support these measures is an interesting future work.

Robust training focuses on training accurate models against noisy or adversarial data. While data can be problematic for many reasons [Xiao et al., 2015], most of the robust training literature assumes label noise [Song et al., 2020] and mitigates it by using (1) model architectures that are resilient against the noise [Chen and Gupta, 2015, Jindal et al., 2016, Bekker and Goldberger, 2016, Han et al., 2018a], (2) loss regularization techniques [Goodfellow et al., 2015, Pereyra et al., 2017, Tanno et al., 2019, Hendrycks et al., 2019, Menon et al., 2020], and (3) loss correction techniques [Patrini et al., 2017, Chang et al., 2017, Ma et al., 2018, Arazo et al., 2019]. However, these works typically do not focus on improving fairness.

A recent trend is to improve both fairness and robustness holistically [Lee et al., 2021]. The closest work to our algorithm is FR-Train [Roh et al., 2020], which performs adversarial training among a classifier, a fairness discriminator, and a robustness discriminator. In addition, FR-Train relies on a clean validation set for robust training. In comparison, our algorithm is a sample selection method that does not require modifying the internals of model training and does not rely on a validation set either. There are other techniques that solve different problems involving fairness and robustness. A recent study observes that feature selection for robustness may lead to unfairness [Khani and Liang, 2021], and another study shows the limits of fair learning under data corruption [Konstantinov and Lampert, 2021]. In addition, there are techniques for (1) fair training against noisy or missing sensitive group information [Hashimoto et al., 2018, Lamy et al., 2019, Lahoti et al., 2020, Wang et al., 2020, Awasthi et al., 2020, Celis et al., 2021], (2) fair training with distributional robustness [Mandal et al., 2020, Rezaei et al., 2021], (3) robust fair training under sample selection bias [Du and Wu, 2021], and (4) fairness-reducing attacks [Chang et al., 2020, Solans et al., 2020]. In comparison, we treat fairness and robustness as equals and improve them together.

**Sample Selection**   While sample selection is traditionally used for robust training, it is recently being used for fair training as well. The techniques are quite different where robust training focuses

on selecting or fixing samples based on their loss values [Han et al., 2018b, Jiang et al., 2018, Chen et al., 2019, Shen and Sanghavi, 2019], while fair training is more about balancing the sample sizes among sensitive groups to avoid discrimination [Roh et al., 2021]. In comparison, we combine the two approaches by selecting samples while keeping a balance among sensitive groups.

# 7 Conclusion

We proposed to our knowledge the first sample selection-based algorithm for fair and robust training. A key formulation is the combinatorial optimization problem of unbiased sampling in the presence of data corruption. We showed that this problem is strongly NP-hard and proposed an efficient and effective algorithm. In the experiments, our algorithm shows better performances in accuracy and fairness compared to other baselines. In comparison to the state-of-the-art FR-Train, our algorithm is competitive and easier to deploy without having to modify the model training or rely on a clean validation set.

**Societal Impact & Limitation**   Although we anticipate our research to have a positive societal impact by considering fairness in model training, it may also have some negative impacts. First, our fairness-aware training may result in worse accuracy. Although we believe that fairness will be indispensable in future machine learning systems, we should not unnecessarily sacrifice accuracy. Second, choosing the right fairness measure is challenging, and a poor choice may lead to undesirable results. In applications like criminal justice and finance where fairness issues have been thoroughly discussed, choosing the right measure may be straightforward. On the other hand, there are new applications like marketing where fairness has only recently become an issue. Here one needs to carefully consider the social context to understand what it means to be fair.

We also discuss limitations. Fairness is a broad concept, and our contributions are currently limited to prominent group fairness measures. There are other important notions of fairness like individual fairness, and we believe that extending our work to address them is an interesting future work. In addition, our robust training is currently limited to addressing label noise. However, ITLM is also known to handle adversarial perturbation on the features, so we believe our algorithm can be extended to other types of attacks as well. Another avenue of research is to use other robust machine learning algorithms like Ren et al. [2018] in addition to ITLM.

# Acknowledgements

Yuji Roh and Steven E. Whang were supported by a Google AI Focused Research Award and by the Engineering Research Center Program by the National Research Foundation of Korea (NRF) grant funded by the Korea government (MSIT) (No. NRF-2018R1A5A1059921 and NRF-2021R1C1C1005999). Kangwook Lee was supported by NSF/Intel Partnership on Machine Learning for Wireless Networking Program under Grant No. CNS-2003129 and by the Understanding and Reducing Inequalities Initiative of the University of Wisconsin-Madison, Office of the Vice Chancellor for Research and Graduate Education with funding from the Wisconsin Alumni Research Foundation. Changho Suh was supported by Institute for Information & communications Technology Planning & Evaluation (IITP) grant funded by the Korea government (MSIT) (No. 2019-0-01396, Development of framework for analyzing, detecting, mitigating of bias in AI model and training data).

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
