# OpenReview forum: "Sample Selection for Fair and Robust Training"
_NeurIPS.cc/2021/Conference — NeurIPS 2021 Poster_

### Official Review · Reviewer_GddM · 2021-07-05

**Rating:** 6
**Confidence:** 3

**Summary:**

This paper proposes a sample selection-based method for training a robust and fair model under data corruption (label flipping). At a high-level, the proposed method combines the essence of ITLM (sample selection method for robustness) and FairBatch (sample selection method for fairness) in a clever way. The method involves three optimizations: (1)minimizing the loss of the selected samples while satisfying fairness constraints; (2)training a model on samples selected from (1); (3) adjusting sampling ratios of each subgroup.
For evaluation, the author compares the proposed method with the state-of-the-art baseline method FR-Train as well as variations of the proposed method on one synthetic dataset and two widely used datasets.

**Limitations And Societal Impact:**

Yes

**Main Review:**

Pros:
- well-written
- comprehensive empirical comparison with baselines and variations
- decent performance without the usage of extra data

Cons:
- the proposed method is inferior to FR-Train when clean validation data is used for FR-Train


The paper is fairly well-written and proposes a method that cleverly combining the essence of two existing methods for robust and fair training. It also conducts an in-depth comparison with the baseline FR-Train and shows that the proposed method can achieve on par results when the extra validation data used for FR-Train is noisy. My remaining concern is that will the proposed method performs better if, say, the extra data is also used? Or alternatively what if the validation data is not given to FR-Train? For example, a subset of the training data can be held out for validation purpose so no extra data advantage will be given to FR-Train to compare with the proposed method.


Question:
- If no extra data is used for FR-Train or extra data is given to the proposed method, will the proposed method outperforms FR-Train?
- How is the proposed method compared with FR-Train in terms of computational cost?


**Time Spent Reviewing:**

2.5

---

> ### Author Response · Authors · 2021-08-10
> **Response to Reviewer GddM**
>
> We thank you for the insightful comments, which we will address in our revision.
>
> Q1. Performance comparison with FR-Train
>
> A1.
> The answer to your question is yes, and we actually have the relevant experiments in the paper. In Table 3, "FR-Train with noisy val. set" is when FR-Train does not have additional clean data and uses a subset of the (noisy) training set as the validation set. As a result, the accuracy and fairness both worsen compared to our algorithm. We will add clarifications in the paper so that readers can more easily find this result.
>
>
> Q2. Computational costs of FR-Train and our algorithm
>
> A2.
> As per your great comment, we measured the runtimes (i.e., wall clock times in seconds) of FR-Train and our algorithm on the synthetic and COMPAS datasets w.r.t. equalized odds:
>
> \--------------------------------------------------------------\
>  $\ \ \ \ \ \ \ \ \ \ \ \ \ \ \ \ \ \  $| $\ \ \ $Synthetic $\ \ \ $| $\ \ \ $COMPAS $\ \ \ $| \
> --------------------------------------------------------------\
> Ours  $\ \ \ \ \ \ \ \ \ \$| $\ \ \ \ \ $44.746  $\ \ \ \ \ $| $\ \ \ \ $ 241.291$\ \ \ \ \ $| \
> FR-Train  $\ \ \ \ \$| $\ \ \ \ \ $90.986 $\ \ \ \ \ $| $\ \ \ \ $ 230.304$\ \ \ \ \ $| \
> \--------------------------------------------------------------
>
> As a result, our algorithm is faster on the synthetic dataset, but a bit slower on the COMPAS dataset. Our algorithm's runtime bottleneck is the greedy algorithm (Algorithm 1) for batch selection. FR-Train trains slowly because it trains three models (one classifier and two discriminators) together. Hence, there is no clear winner, and we will add this comparison in our revision.

---

### Official Review · Reviewer_75jQ · 2021-07-16

**Rating:** 6
**Confidence:** 3

**Summary:**

The paper presents a technique for subsampling training data in order to simultaneously ensure fairness with respect to protected characteristics and robustness in the face of corrupted data. As formulated the problem is NP-hard, so the authors devise a greedy heuristic for solving it and show that applying the technique to logistic regression gives competitive empirical results.

**Limitations And Societal Impact:**

The authors responsibly discuss the limitations of quantitative fairness definitions and the dangers of carelessly applying them in settings which might not make sense.

**Main Review:**

There are common techniques for subsampling datasets to improve performance, but they may actually make unfairness worse by removing samples in a way related to protected characteristics. So the idea here is to subsample a dataset while constraining the choice to ensure that no (protected attribute, label) group is sampled more than some thresholds (which are set adaptively during the training loop).

The general idea is then that training with subsampling of data given these fairness constraints represents a bilevel optimization problem -- the top level is the model minimizing loss, and the bottom level is subsampling in order to maximize the top level. (Although both are minimizing almost the same loss, so it might seem like it's just a single joint minimization problem, the loss the model actually trains on is reweighted in such a way that the problem is general sum, so I think it really does need to be thought of as bilevel.)

This problem is then converted into a knapsack problem, which in turn can be approximately solved using a greedy heuristic. Experiments are run on synthetic datasets and COMPAS, with datasets corrupted by different types of label flipping.

Originality:

The paper mostly combines existing approaches and techniques, but to solve a new problem which it convincingly shows has not been previously addressed.

Clarity:

The paper is clearly written, there are no issues with presentation.

Quality:

The experiments make reasonable choices and are fairly convincing and thorough, although on only one non-synthetic dataset. The definitions of fairness are reasonable choices from the literature.

Significance:

The technique is a solution to a reasonable but pretty narrow problem: when one wants to improve performance using subsampling while also guaranteeing fairness constraints.

Update after rebuttal: the authors have convinced me that this combination of fairness/robustness due to dataset corruption is of more general interest than I had first thought.


**Time Spent Reviewing:**

3.5

---

> ### Author Response · Authors · 2021-08-10
> **Response to Reviewer 75jQ**
>
> We thank you for the insightful comments, which we will address in our revision.
>
> Q1. Quality - Experiments
>
> A1.
> We appreciate your positive comments. One clarification is that we do experiment on two non-synthetic datasets: COMPAS and AdultCensus. We show the AdultCensus results in the supplementary (Table 6) because they are similar to the COMPAS results.
>
>
> Q2. Significance
>
> A2.
> We appreciate the viewpoint and would like to explain why we believe fair and robust training is becoming important. Traditionally, fair training and robust training have been studied separately, but a recent trend is to address them holistically (e.g., [1]) while obtaining the best of both worlds. Hence, this field is nascent, but growing rapidly. Our key contribution is to use sampling as a tool for attaining both fairness and robustness.
>
> [1] Lee et al., "Machine Learning Robustness, Fairness, and their Convergence", SIGKDD 2021

---

### Official Review · Reviewer_pcux · 2021-07-16

**Rating:** 6
**Confidence:** 4

**Summary:**

This paper deals with unfairness in the clean sample selection method, a method to make the learned model robust against the outliers. The authors develop a mechanism to enforce fairness to the clean sample selection method. Their method is formulated as a combinatorial optimization problem; however, the optimization problem is strongly NP-hard and is difficult to solve. To overcome this difficulty, they introduce a greedy algorithm that works well in practice. The empirical evaluations demonstrate that their method outperforms the baselines and approaches to the ideal performance.

**Limitations And Societal Impact:**

The limitations and potential societal impact are adequately addressed.

**Main Review:**

This paper introduces some interesting ideas for fair and robust learning. The results are surprising and interesting; even if we employ the fair classification method, the fairness can be worsened by applying the clean sample selection method. The empirical results demonstrate that the present method overcomes such an issue. Hence, my current decision is acceptance.

The design of the optimization problem of Eq. 1 and Eq. 2 is unclear for me. Specifically, I'm wondering why the second constraint in Eq. 2 is upper bounds on the sample sizes and is not lower bounds. The lower bound constraints of the sample sizes are understandable because if there is an extremely small group, the estimation error of the fairness measure can be large. Hence, the lower bound constraints are justified as a requirement for accurate estimation of the fairness score. However, I cannot come up with a justification for the upper bound constraint.

The parameter choices of the existing fair learning algorithm are not provided. The compared methods, e.g., the method from Zafar et al. 2017, have a parameter that controls the trade-off between fairness and accuracy. The choice of the trade-off parameter highly influences the fairness performance of these methods. I concern that comparison might be unfair if the parameter choice is inappropriate.

**Time Spent Reviewing:**

7

---

> ### Author Response · Authors · 2021-08-10
> **Response to Reviewer pcux**
>
> We thank you for the insightful comments, which we will address in our revision.
>
> Q1. Design of optimization problem
>
> A1.
> We would like to clarify the role of the upper bounds in the second constraint of Eq.2. In principle, we would like to specify equality constraints to determine the sampling rates of different groups for fair sample selection. However, our solution may be infeasible under the equality constraints, so we specify upper bounds instead. Here the upper bounds are adaptively adjusted using techniques in FairBatch (Section 3.3).
>
> We note that lower bounds are not necessary because, when we measure fairness, we use the entire training set instead of the selected set. That is, each group size in the selected set does not affect the fairness estimator error.
>
> We will add these points in our revision.
>
>
> Q2. Hyperparameter choices of the baselines
>
> A2.
> We will explain the hyperparameter choices of baselines in our revision. To make a fair comparison, we aligned the accuracy values as much as possible and then compared the fairness. Specifically, for all baselines, we started from a candidate set of hyperparameters (e.g., a range [0.1, 10] for fairness tuning knob in Zafar et al. 2017) and used cross-validation to choose the hyperparameters that result in the best fairness while having an accuracy that best aligns with other results.

---

### Official Review · Reviewer_5Ear · 2021-07-17

**Rating:** 6
**Confidence:** 4

**Summary:**

The paper introduces an algorithm for training models that are both fair and robust to noise in the data. A number of experiments shows the effectiveness of the method for improving both fairness and robustness.

**Limitations And Societal Impact:**

None that I can foresee.

**Main Review:**

$\textbf{Originality}$

To my awareness the presented algorithm is novel and studying practical algorithms for designing models that are both robust and fair is an underexplored area. The proposed algorithm consists of multiple steps that are related to prior methods, which is properly acknowledged in the text.

Regarding related work, I find that a few recent papers that deal with the interplay between robust and fair ML can be discussed in addition to those presented in the paper. In particular, [1] studies the limits of fair learning under data corruption and [2,3,4,5] are additional recent works that study fairness under label/attribute noise.

$\textbf{Quality}$

The work appears technically sound, with the algorithm and the experimental setups discussed in sufficient detail.

$\textbf{Clarity}$

Overall, the paper is well-written and easy to follow. As a minor suggestion, on page 3, equation (2) can include $S_y$ written explicitly in terms of the variables $p_j$ to improve clarity.

$\textbf{Significance}$

Designing algorithms that improve both fairness and robustness at the same time is certainly important. The presented method seems to be relatively effective in achieving this, although the performance appears relatively inconsistent between datasets. Nevertheless, since there is little prior work on the topic, I still find the presented results to be useful and of interests. Future work can compare this algorithm to other baselines for example running Iterative Trimmed  Loss  Minimization or another robust ML algorithm on a fairness-regularized objective.


$\textbf{References}$

[1] N. Konstantinov and C. H. Lampert. Fairness-aware learning from corrupted data, ArXiv, 2021

[2] A. L. Lamy et al. Noise-tolerant fair classification. NeurIPS, 2019.

[3] P. Awasthi, M. Kleindessner and J. Morgenstern. Equalized odds postprocessing under imperfect group information. In: AISTATS, 2020.

[4] A. Mehrotra and L. E. Celis. Mitigating Bias in Set Selection with Noisy Protected Attributes. In: Conference on Fairness, Accountability, and Transparency, 2021.

[5] L. Celis, L. Elisa, L. Huang and N. K. Vishnoi. Fair Classification with Noisy Protected Attributes. ArXiv, 2020

**Time Spent Reviewing:**

5

---

> ### Author Response · Authors · 2021-08-10
> **Response to Reviewer 5Ear**
>
> We thank you for the insightful comments, which we will address in our revision.
>
>
> Q1. Related work
>
> A1.
> We appreciate the papers and agree that they propose important approaches for fair and robust training. As you mentioned, paper [1] shows that data corruption can inevitably hurt fair training and that the quality of training data is thus very critical. In addition, papers [2,3,4,5] aim to make fair training more robust when the sensitive attribute is noisy, a scenario we would like to consider in the future. We will cite the papers and add discussions in our revision.
>
>
> Q2. Future work
>
> A2.
> We thank you for suggesting future work. So far, we did observe that combining Iterative Trimmed Loss Minimization (ITLM) with a fairness-regularized method (called Penalty in our paper) [Zafar et al., 2017] usually shows worse accuracy and fairness than our algorithm, as demonstrated by the baselines ITLM->Penalty (Tables 1, 2, 5, and 6) and (ITLM+Penalty)->Penalty (Table 5)*. For example in Table 1, when we train the algorithms for equalized odds under label flipping attack, the (accuracy, unfairness) result of our algorithm is (0.727, 0.064), while the result of ITLM->Penalty is (0.651, 0.172). In addition, we agree it would be interesting to also use a robust ML algorithm (e.g., Ren et al., ICML 2018) other than ITML and will add a discussion in our revision.
>
> *Tables 5 and 6 are in the supplementary.
>
>
>
> Q3. A minor comment -- Clarifying an expression
>
> A3.
> We will clarify $S_y$ in Eq. (2) by writing it in terms of $p_j$ as follows: $|S_y| = \sum_{j \in \mathbb{I}_{y}} p_j$.

---

### Decision · Program_Chairs · 2021-09-28

**Decision:**

Accept (Poster)

**Comment:**

Reviewers were unanimous in appreciating the paper's contribution to a topical problem, namely ensuring model fairness and robustness. The motivating analysis of the potential adverse effects of clean-sample selection for fairness were seen as particularly interesting. The reviews did bring up some areas for potential improvement, including citing a broader line of work on fairness under corruption, and clarifying some technical details in the algorithm. The authors are encouraged to incorporate these for an update version of the paper.

**Consistency Experiment:**

NeurIPS has a long history of experimentation. In 2014, NeurIPS ran an experiment in which 10% of submissions were reviewed by two independent committees to quantify the randomness in the review process. This year, we repeated a variant of this experiment to see how the quality of the review process has changed over time.  This paper was part of the experiment and was therefore assigned to two committees (consisting of reviewers, an Area Chair, and a Senior Area Chair) that reached independent decisions.  If both committees made the same recommendation, this recommendation was followed. If a single committee recommended acceptance, the paper was accepted (with the exception of a few cases in which the other committee identified what we considered a fatal flaw, e.g., an error in a key result).

This copy’s committee reached the following decision: **Accept (Poster)**

The other committee assigned to the paper recommended **Reject**.  You can find the other set of reviews, along with any follow up discussion with the authors here:
https://openreview.net/forum?id=IZNR0RDtGp3